# Adsorption Removal Characteristics of Hazardous Metalloids (Antimony and Arsenic) According to Their Ionic Properties

**Seung-Hun Lee** [1], **Jinwook Chung** [2] and **Yong-Woo Lee** [1,*]

1 Department of Chemical and Molecular Engineering, Hanyang University, 55 Hanyangdaehak-Ro, Sangrok-Gu, Ansan 15588, Gyeonggi-Do, Republic of Korea; lck2394sin@hanyang.ac.kr
2 R&D Center, Samsung Engineering Co., Ltd., 41 Maeyoung-Ro, 269 Beon-Gil, Youngtong-Gu, Suwon 16523, Gyeonggi-Do, Republic of Korea; chung.j@samsung.com
* Correspondence: yongwoolee@hanyang.ac.kr

**Abstract:** Antimony and arsenic, which have a high carcinogenicity, should be removed depending on their ionic charge in water. Therefore, we attempted to confirm the adsorption characteristics of antimony and arsenic considering ionic charge to improve removal efficiency. We used palm-based activated carbon (PAC), coal-based activated carbon (CAC), modified activated carbon (MAC), styrene-divinylbenzene copolymer (SP825), activated alumina (AA), and zeolite as adsorbents for antimony and arsenic. Negatively charged adsorbents (CAC, PAC, MAC, and zeolite) with similar zeta potentials showed better removal efficiency as the surface area increased. However, SP825, which is almost neutral, and AA, which is positively charged, exhibited a high removal efficiency (100%) for arsenic and Sb(V), which are anions, regardless of surface area. However, due to the price, coal-based activated carbon or palm-based activated carbon is considered more advantageous than using AA or SP825. Last, during the arsenic adsorption process, As(III) was oxidized to As(V) due to Fe(II) contained in activated carbon. The addition of activated carbon can improve oxidation efficiencies of As(III) before coagulation and precipitation, in which As(V) is easier to remove than As(III).

**Keywords:** absorption; antimony; arsenic; ion properties; metalloid

## 1. Introduction

The number and type of hazardous chemicals discharged into water systems have increased over time. Among these hazardous chemicals, six—boron, silicon, germanium, antimony, arsenic, and tellurium—have intermediate properties between metals and non-metals and are called metalloids. Of these metalloids, antimony and arsenic are hazardous and are under regulation of emission and exposure standards in many countries [1–3]. The World Health Organization (WHO) standardized 0.02 mg/L as a safe antimony concentration in drinking water, and the US Environmental Protection Agency (EPA) set 0.006 mg/L as a standard, while the European Union (EU) set a standard of 0.005 mg/L. The WHO, EPA, and EU all selected and manage a standard of 0.01 mg/L for arsenic [4–6].

Antimony is commonly used as a flame retardant and is generated in smelting of nonferrous metals including copper or gold, mining, leaching from plastics and waste incineration [7,8].

Antimony is an irritant to mucous membranes, eyes, and skin. Antimony(III) oxide and antimony(V) oxide damage the lungs, and antimony sulfide is a cardiac poison. The International Agency for Research on Cancer (IARC) classified antimony as a group 2B (possibly carcinogenic), and the American Conference of Governmental Industrial Hygienists (ACGIH) classified it as A2 (suspected human carcinogen) [1,6,8].

Arsenic, which is mainly used as an alloying additive or rodenticide, is generated in the smelting of nonferrous metal including copper or gold and in producing sulfur-based products [4,9]. Arsenic is a human group I carcinogen in the IARC carcinogenicity classification and causes lung cancer, skin cancer, liver cancer, and bladder cancer in humans. Arsenic has gained extensive attention as a chemical with great risk to the

human body, as it can have both carcinogenic effects and various non-carcinogenic toxic effects, such as digestive disorders, skin diseases, nervous system diseases, diabetes, liver disease, anemia, hematological diseases, eye diseases, respiratory system diseases, and cardiovascular diseases [10].

Though removal studies on antimony and arsenic have progressed considerably, the two chemicals do not exist as single ions in water. Therefore, removal studies that did not consider ionic charge have limitations in understanding the removal characteristics. For example, Zhu's study on electrocoagulation removal of antimony published in 2011 used atomic fluorescence spectrometry (AFS), and Liu's study on antimony coagulation removal in 2021 used AFS and inductive coupled plasma (ICP) [11]. Additionally, a study on the adsorption and removal of arsenic published in 2021 used only atomic absorption (AA) [8,10,12]. Sb(III) and Sb(V) ions exist as $Sb(OH)_3$ and $Sb(OH)_6{}^-$, respectively, at pH 7, while arsenic exists in a total of eight neutral molecules or anions (i.e., $H_3AsO_3$, $H_3AsO_4$, $H_2AsO_3{}^-$, $H_2AsO_4{}^-$, $HAsO_3{}^{2-}$, $HAsO_4{}^{2-}$, $AsO_3{}^{3-}$, $AsO_4{}^{3-}$) in water depending on temperature and pH [2,13,14]. The unique behavior of each ion in water must be considered during analysis and processing [4,9,15]. Additionally, the most commonly used technologies for removal of metalloids are coagulation and precipitation, and adsorption is performed downstream to maximize treatment efficiency [5]. Therefore, in the present study, the adsorption and removal characteristics of arsenic and antimony were confirmed based on the ionic charge in water.

## 2. Materials and Methods

### 2.1. Chemicals

The Sb(III) used in the study was $K_2Sb_2(C_4H_2O_6)_2$ (Duksan Reagents, Extra pure grade, Ansan-si, Gyeonggi-Do, Republic of Korea), Sb(V) was $KSb(OH)_6$ (Sigma–Aldrich, St. Louis, MO, USA), As(III) was $As_2O_3$ (Sigma–Aldrich, ReagentPlus®, St. Louis, MO, USA), and As(V) was $Na_2HAsO_4 \cdot 7H_2O$ (Sigma–Aldrich, ≧98.0%, St. Louis, MO, USA). As reagents for analysis, HCl (Fluka, Trace Select®, Charlotte, North Carolina, Switzerland), L-ascorbic acid (Sigma–Aldrich, ACS reagent, St. Louis, MO, USA), and KOH (Sigma–Aldrich, 99.99% trace metals basis, St. Louis, MO, USA) were used. As and Sb calibration curves were prepared from 20 mg/L As(III) standard solution and Sb(III) standard solution (Modern Water, standard solution, New Castle, DE, USA).

### 2.2. Hazardous Metalloid Analysis Methods

Antimony and arsenic have different processing characteristics depending on their form in water. Anodic stripping voltammetry (ASV, Modern Water, PDV 6000 plus, London, UK) was used to measure the ion concentration of hazardous metalloids. First, for antimony, total Sb and Sb(III) were measured using a mercury film electrode. Sb(V) was calculated by subtracting the measured value of Sb(III) from the total Sb measured value. Three percent hydrochloric acid (Modern Water, Antimony(III) Electrolyte A, London, UK) was used as the electrolyte, in which 7 g of ascorbic acid (Modern Water, Antimony(III) Electrolyte B, New Castle, DE, USA) was dissolved to prevent changes in the concentration of Sb(III) due to pH of the electrolyte. Sb(III) and total Sb were measured by creating a calibration curve within the ranges of 0.005–0.100 mg/L and 0.005–0.100 mg/L, respectively. For arsenic, total As and As(III) were measured using a solid gold electrode and a gold film electrode. As(V) was calculated by subtracting the measured value of As(III) from the total As measured value. Four percent nitric acid (Modern Water Electrolyte Diluent B, New Castle, DE, USA) was used as the electrolyte, in which 7% acetic acid (Modern Water As Electrolyte concentrate, New Castle, DE, USA) and 7 g of ascorbic acid (Modern Water As(III) Electrolyte B, New Castle, DE, USA) were dissolved. A calibration curve was prepared in the range of 0.005–0.100 mg/L for As(III) and 0.025–0.200 mg/L In addition, the total metalloid concentration was analyzed using inductively coupled plasma–mass spectroscopy (ICP-MS, Perkin Elmer, NexION 300X, Waltham, MA, USA) and was compared with the total Sb and As results using ASV.

### 2.3. Batch Tests

For the batch test of the adsorption treatment, commonly reported adsorbents were chosen for each metalloid. For antimony, palm-based activated carbon (PAC, Jeilcarbontec, 0.6–2.4 mm, Goyang-si, Gyeonggi-do, Republic of Korea) was selected as a commonly used commercial adsorbent, and SP825 (Mitsubishi Chemical Corporation, styrene di­vinylbenzene copolymer, Tokyo, Japan) was selected considering zeta potential. Zeolite (Dongsin, 0.6–2.4 mm, Gyeongju-si, Gyeongsangbuk-do, Republic of Korea) had a neutral zeta potential was chosen as the control due to its smallest least adsorption effect. More­over, MAC was made by reacting a previous PAC for 120 min at 900 °C while flowing 1000 mL/min of $CO_2$ gas to improve adsorption efficiency [5,14]. For arsenic, coal-based activated carbon (CAC, Jeilcarbontec, 0.6–2.4 mm, Goyang-si, Gyeonggi-do, Republic of Korea) was selected as an adsorbent to replace the commercial adsorbents PAC and MAC. Activated alumina (AA, BASF, 2 mm, Ludwigshafen, Germany) had a positive zeta poten­tial was selected considering zeta potential, and zeolite (Dongsin, 0.6–2.4 mm, Gyeongju-si, Gyeongsangbuk-do, Republic of Korea) was selected as a control adsorbent [5,14,16]. If the diameter of the adsorbent was small, the surface area increased, and the contaminant removal efficiency increased. However, regardless of removal efficiency, the water flow rate inside the adsorption tower decreases, and the water head may be lost for small diameters of the adsorbent, reducing the overall treatment efficiency. Considering this, the diameter range was 0.6–2.4 mm, which is mainly used for water processing. The physical properties provided by the manufacturer of the selected adsorbent are summarized in Table 1. Before the experiment, the specific surface area of each adsorbent was analyzed using a Brunauer Emmett Teller (BET) Analyzer (MicrotracBEL Corp., BELSORP MAX X, Osaka, Japan). Also, the zeta potential was analyzed with a zeta potential analyzer (Brookhaven Instruments Corporation, NanoBrook ZetaPALS Potential Analyzer, Holtsville, NY, USA) at pH 7.

**Table 1.** Absorbents characteristics.

| Category | PAC | MAC | CAC | AA | Zeolite | SP825 |
|---|---|---|---|---|---|---|
| Surface area ($m^2/g$) | 1100 | 1904 | 1050 | 360 | <800 | 977 |
| Density ($g/cm^3$) | 1.21 | 1.16 | 2.10 | 3.97 | 2.37 | 1.01 |
| Size (mm) | 0.6–2.4 | 0.6–2.4 | 0.6–2.4 | 2.0 | 1–3 | 0.2–1.2 |

In the case of antimony, 30 mL of Sb(III) and Sb(V) solutions with a concentration of 0.2 mg/L each were added to the conical tube, and 20 mg of each adsorbent was added. Afterwards, it was stirred for 10, 30, 60, 120, and 240 min and then filtered to collect samples.

For arsenic, 3 g of the adsorbent was soaked in distilled water to remove air in the pores one day before the experiment. The air in the adsorbent pores was removed to solve the problem of the adsorbent floating during the experiment. The soaked adsorbent was added to 1 L of a 10 mg/L As solution and then stirred using a jar tester (C-JY-H, Vision Lab Science, Incheon-si, Republic of Korea) at 120 rpm for 1440 min to ensure complete mixing of the adsorbent and solution. The ratios between As(III) and As(V) were selected as 2:0, 1:1, and 0:2, with a total concentration of 10 mg/L, so that the removal efficiency of each ion was found. Samples were collected and analyzed at regular time intervals to observe the reaching of adsorption equilibrium.

### 2.4. Continuous Tests

Based on the batch test results, a continuous adsorption experiment was performed to design an adsorption tower for wastewater treatment to remove metalloids. The lab-scale continuous adsorption reactor consisted of five acrylic columns with an inner diameter of 1.5 cm, a height of 1.7 m, and an internal volume of 300.3 mL, and three constant volume transfer pumps (Plus master, PP-150, Daejeon, Republic of Korea). Two 200 L raw water tanks were installed to stably supply the set flow rate to each pipe. Additionally, before the

operation, the sample flowed for 24 h to remove fine particles of adsorbent and to wet the inside of the micropores.

To determine the antimony removal characteristics, the flow rate experiment ranged from linear velocity (LV) 5, which injects 21.2 L/day of raw water per pipe, to LV 1, which injects 4.2 L/day of raw water, and the experiment was conducted for 6 days, allotting 1 day for stabilization and 5 days for actual operation of each condition. LV means line unit speed, and its unit is m/h. The bed height was 33 cm, and the surface velocity (unit $h^{-1}$) corresponds to space velocity (SV) 15 for LV 5, SV 9 for LV 3, and SV 3 for LV 1. Each column was divided into PAC, MAC, zeolite, SP825, and control columns and filled with adsorbent. The initial injection concentration was set to 0.2 mg/L, and Sb(III):Sb(V) = 1:1. The inflow rate of raw water was selected at 4.2 L/day of LV 1, which was the slowest flow rate to increase the contact time with activated carbon. Afterwards, the amount of adsorbent filling was changed to increase the contact time. At this time, the change in filling amount started from 33 cm (LV 1, SV 3), which is the adsorbent height in the existing flow rate change test, and was applied up to 100 cm (LV 1, SV 1). The contact time was 5 min at LV 5, 10 min at LV 3, 15 min at LV 1 (SV 3), 30 min at SV 2, and 60 min at SV 1.

The flow rate experiment for arsenic was conducted at LV 4, which injects 17.0 L/day of raw water to LV 1, which injects 4.2 L/day of raw water. Columns were divided into CAC, PAC, AA, zeolite, and control columns and were filled with adsorbent. The initial injection concentration was set to 2 mg/L based on the batch test results, and As(III):As(V) = 1:1. As with antimony, the inflow rate of raw water was 4.2 L/day at LV 1, which was the slowest flow rate in the experiment to increase the contact time with activated carbon. The adsorbent filling amount was changed to the same conditions as for antimony.

## 3. Results and Discussion

### 3.1. Metalloids Analysis Using ASV

We used ASV to confirm the behavior characteristics of each ion. Previous studies indicated no known existing method for analyzing total Sb using ASV. Thus, before researching the removal characteristics of antimony, it is vital to set up an analysis method using ASV. Considering the form of Sb(V) in water depending on pH, previous research results showed that Sb(V) exists in the form of Sb(III) in a highly concentrated HCl solution (5 M or more) [17]. However, if the sample is preprocessed with HCl, the mercury film coated on the working electrode of the ASV dissolves, preventing measurements. To prevent the mercury film from dissolving, the sample preprocessed with a 5 M HCl solution was re-neutralized into a basic solution. However, as the pH became neutral, Sb(III) was oxidized back to Sb(V). These results demonstrate that Sb(III) is unstable in water and is easily oxidized to Sb(V); when organic acid is added to the solution, Sb(III) is maintained without oxidization [18], L-ascorbic acid, which is used as an electrolyte for ASV, was added for preprocessing. For the final preprocessing, 45 mL of 5 M HCl and 1.4 g of L-ascorbic acid were added to 10 mL of the sample and left for 10 min, and 45 mL of 5 M KOH was added here and cooled to room temperature. After preprocessing, mercury film dissolution of the working electrode did not occur, and a trend line at the level of $R^2 = 0.9969$ was stably obtained Sb(III), As(III), and total As were analyzed using the measurement method distributed by the ASV manufacturer, and the $R^2$ was Sb(III): 0.9693, As(III): 0.9980, and total As: was found to be 0.9925 (Figure 1).

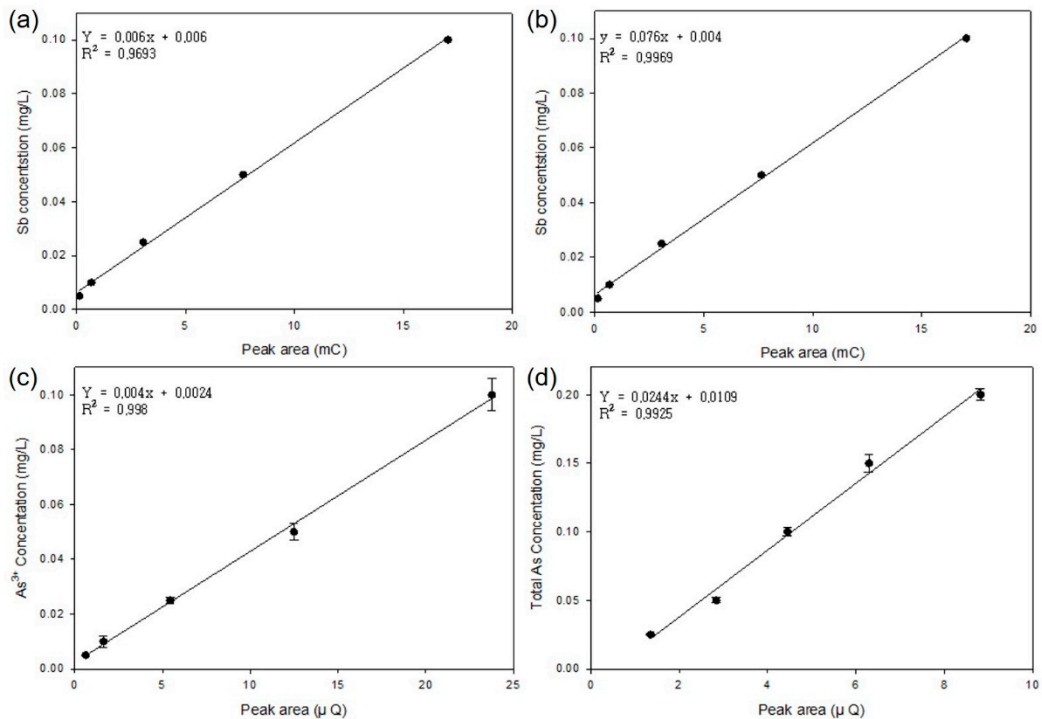

**Figure 1.** Calibration curve ((**a**) Sb(III), (**b**) Total Sb, (**c**) As(III), (**d**) Total As).

*3.2. Metalloid Adsorption Removal Characteristics*

To interpret the antimony adsorption processing results, the surface areas of the selected adsorbents, PAC, zeolite, SP825 made of styrene-divinylbenzene copolymer, and MAC were analyzed using a BET analyzer. Consequently, the specific surface area of PAC was 1098 m$^2$/g, zeolite was 678 m$^2$/g, SP825 was 977 m$^2$/g, and MAC was 1904 m$^2$/g.

To find the adsorption characteristics of PAC, MAC, zeolite, and SP825, the maximum adsorption amount was determined through an isotherm study. After writing and comparing the Langmuir adsorption isotherm (1) and the Freundlich equation (2), a more appropriate adsorption isotherm was determined. First, the Langmuir isotherm had a stable linear equation, with a R$^2$ value of the adsorption isotherm greater than 0.96 for all antimony ions. In contrast, the Freundlich isotherm showed linearity with an R$^2$ value of about 0.94 and was not suitable. Accordingly, it was more appropriate to apply the Langmuir isotherm rather than the Freundlich isotherm for adsorption removal of antimony (see Table 2). The $q_m$ values for Sb(III) were 0.126 mg/g for PAC, 0.159 mg/g for MAC, 0.056 mg/g for zeolite, and 0.035 mg/g for SP825. The $q_m$ value for Sb(V) was 0.062 mg/g for PAC, 0.090 mg/g for MAC, 0.038 mg/g for zeolite, and 0.118 mg/g for SP825. They are listed in order of size as follows:

- Maximum adsorption amount for Sb(III) ($q_m$): MAC >PAC > Zeolite > SP825
- Maximum adsorption amount for Sb(V) ($q_m$): SP825 > MAC > PAC > Zeolite

$$q_e = -\frac{1}{b}\frac{q_e}{C_e} + q_m \tag{1}$$

$$\log Q_e = \frac{1}{n}\log C_e + \log K_f \tag{2}$$

**Table 2.** Langmuir and Freundlich isotherm parameters for Sb.

| Category | | Langmuir | | | Freundlich | | |
|---|---|---|---|---|---|---|---|
| | | $q_m$ | $-\frac{1}{b}\frac{q_e}{C_e}$ | $R^2$ | $q_m$ | $\frac{1}{n}$ | $R^2$ |
| Sb(III) | PAC | 0.126 | 0.1121 | 0.9843 | 0.124 | 0.0482 | 0.9513 |
| | MAC | 0.159 | 0.0895 | 0.9673 | 0.155 | 0.0298 | 0.9147 |
| | Zeolite | 0.056 | 0.1562 | 0.9981 | 0.058 | 0.0781 | 0.9468 |
| | SP825 | 0.035 | 0.1701 | 0.9998 | 0.031 | 0.0750 | 0.9678 |
| Sb(V) | PAC | 0.062 | 0.1593 | 0.9975 | 0.058 | 0.0548 | 0.9384 |
| | MAC | 0.090 | 0.1393 | 0.9930 | 0.096 | 0.0487 | 0.9288 |
| | Zeolite | 0.038 | 0.1710 | 0.9996 | 0.033 | 0.0742 | 0.9455 |
| | SP825 | 0.118 | 0.1180 | 0.9863 | 0.129 | 0.0318 | 0.9417 |

Except for SP825, the adsorption performance of the adsorbent per unit mass in an aqueous solution in which Sb(III) is dissolved showed that the removal efficiency of Sb(III) in aqueous solution varied depending on the surface area rather than the surface properties. Sb(III) is electrically neutral as it exists as $Sb(OH)_3$ in water. Therefore, in adsorption removal, removal occurs only by physical adsorption, so MAC with a relatively high surface area showed a high removal rate, and zeolite with the lowest surface area had the lowest removal rate. In contrast, SP825, which has a lower surface area than PAC or MAC, resulted in a higher removal rate for Sb(V). As Sb(V) exists in the form of $Sb(OH)_6{}^-$ in water, factors other than the surface area of the adsorbent play a crucial role in removing Sb(V).

Next, to interpret the arsenic adsorption processing results, the surface areas of the selected adsorbents (CAC, PAC, AA, and zeolite) were analyzed using a BET analyzer. The device showed specific surface area of CAC as 1149.8 $m^2/g$, PAC as 1098 $m^2/g$, AA as 342 $m^2/g$, and zeolite as 678 $m^2/g$.

Table 3 lists the arsenic ion adsorption experiment results for all adsorbents. The adsorption performance of each adsorbent differed and depended on factors other than simple surface area. In particular, in AA, the adsorption efficiency for arsenic is high despite its very small surface area (342 $m^2/g$) compared to other adsorbents. This result is attributed to the effect of zeta potential. Since it has been reported that the zeta potential of AA is positive at pH 9 or lower, arsenic, which is mostly in anionic form in water, is easily adsorbed under the influence of zeta potential [16,19].

**Table 3.** Maximum adsorption of As ions by adsorbent.

| Category | | CAC | PAC | Activated Alumina | Zeolite |
|---|---|---|---|---|---|
| As(III):As(V) 2:0 | Total As (mg/g) | 2.17 | 1.55 | 2.49 | 0.10 |
| | As(III) (mg/g) | 3.58 | 2.61 | 2.55 | 0.50 |
| As(III):As(V) 1:1 | Total As (mg/g) | 3.01 | 1.95 | 2.04 | 0.44 |
| | As(III) (mg/g) | 1.47 | 1.46 | 0.95 | 0.26 |
| | As(V) (mg/g) | 1.53 | 0.49 | 1.08 | 0.18 |
| As(III):As(V) 0:2 | Total As (mg/g) | 1.06 | 0.97 | 2.01 | 0.21 |

In addition, a unique adsorption trend was observed in PAC at a concentration ratio of 1:1 because the adsorption ratio of As(V) temporarily showed a negative value. A previous study reported that the oxidation of As(III) to As(V) is accelerated by iron and manganese oxides present in activated carbon. It is presumed that As(III) is oxidized to As(V) during the adsorption process [20–22]. Also, the removal efficiency of As(III):As(V) = 1:1 was higher than that of As(III):As(V) = 2:0 or As(III):As(V) = 0:2 when changing the maximum adsorption amount of total arsenic due to the arsenic ion ratio in CAC. Considering the previous results showing that Fe(0) is favorable for adsorbing As(V) when As(III) is oxidized to As(V) during the adsorption reaction, iron ions present in activated carbon are reduced to form Fe(0), which is expected to increase the efficiency of arsenic removal [23]. Moreover, As(III):As(V) = 2:0 is expected to be less affected by the increase in Fe(0) as As(V) does not exist initially. For As(III):As(V) = 0:2, As(III) does not exist, so it is expected that iron ions were not reduced to Fe(0). Then, an isothermal adsorption experiment was conducted for each arsenic ion ratio targeting CAC to identify the removal characteristics of each arsenic ion, which was the most efficient at a 1:1 ratio, and the adsorption isotherm equation was prepared (see Table 4). Compared with Langmuir adsorption isotherm (1) and Freundlich adsorption isotherm, a more appropriate adsorption isotherm was determined. In the Langmuir isotherm graph, the $R^2$ value of the adsorption isotherm for all arsenic ion ratios was greater than 0.98, indicating a stable linear trend. In contrast, the Freundlich isotherm obtained a trend line equation with high linearity with a $R^2$ value of 0.99 or more for some parts but was not suitable when the As(III) ratio was low.

**Table 4.** Langmuir and Freundlich isotherm parameters for As.

| As(III):As(V) | Langmuir | | | Freundlich | | |
|---|---|---|---|---|---|---|
| | $q_m$ | $-\frac{1}{b}\frac{q_e}{C_e}$ | $R^2$ | $q_m$ | $\frac{1}{n}$ | $R^2$ |
| 2:0 | 2.39 | 0.4201 | 0.9998 | 2.30 | 0.0244 | 0.9795 |
| 1.5:0.5 | 2.86 | 0.3502 | 0.9997 | 2.74 | 0.0256 | 0.9450 |
| 1:1 | 3.91 | 0.2560 | 0.9859 | 3.60 | 0.0668 | 0.9981 |
| 0.5:1.5 | 4.19 | 0.2384 | 0.9858 | 3.80 | 0.0710 | 0.9957 |
| 0:2 | 4.28 | 0.2336 | 0.9862 | 3.91 | 0.0677 | 0.9995 |

*3.3. Continuous Adsorption Tests*

Figure 2 illustrates that the amount of antimony removed gradually increased as contact time increased. For removal performance for each adsorbent, MAC removed 100% of antimony under the conditions of LV 1 m/h and SV 2 1/h when the adsorbent was filled to 0.5 m. For PAC, up to 92.1% of total antimony was removed under the conditions of LV 1 m/h and SV 1 1/h with the adsorbent filled to 1 m. For SP825, the total antimony removal rate is not that high, at a maximum of 66%, but the removal rate for Sb(V) showed a high removal efficiency of 100% under the conditions of LV 1 m/h and SV 2 1/h. However, when comparing price and performance, MAC and SP825 are more than eight times more expensive than PAC, making it difficult to apply them to actual industrial wastewater treatment. Accordingly, the applicable adsorbent is PAC, and it would be appropriate to operate at LV 1 m/h and SV 1 1/h.

Figure 3 demonstrates that the overall removal efficiency was favorable for AA, but the efficiency of CAC and PAC continued to increase as the contact time increased, reaching the treatment efficiency target under the conditions of LV 1 m/h and SV 2 1/h with a 30-min contact time. Comparing CAC and PAC, the efficiency of PAC was high in a 5- to 20-min contact time with a filling height of 33 cm, but the efficiency of CAC was high during the 30-min contact time with a filling height of 50 cm. This is due to the formation of a detour in the column caused by the short packing height. The removal efficiency of As(III) was advantageous for CAC and PAC, but the efficiency of AA increased as the contact time increased. The high As(III) removal efficiency of these activated carbons is presumed to

be due to the oxidation of As(III) to As(V), which was confirmed in previous batch tests. The removal efficiency of As(V) was favorable compared to that of AA, but the continued increase in efficiency of activated carbon as the contact time increased is presumed to be due to the oxidation of As(III) to As(V).

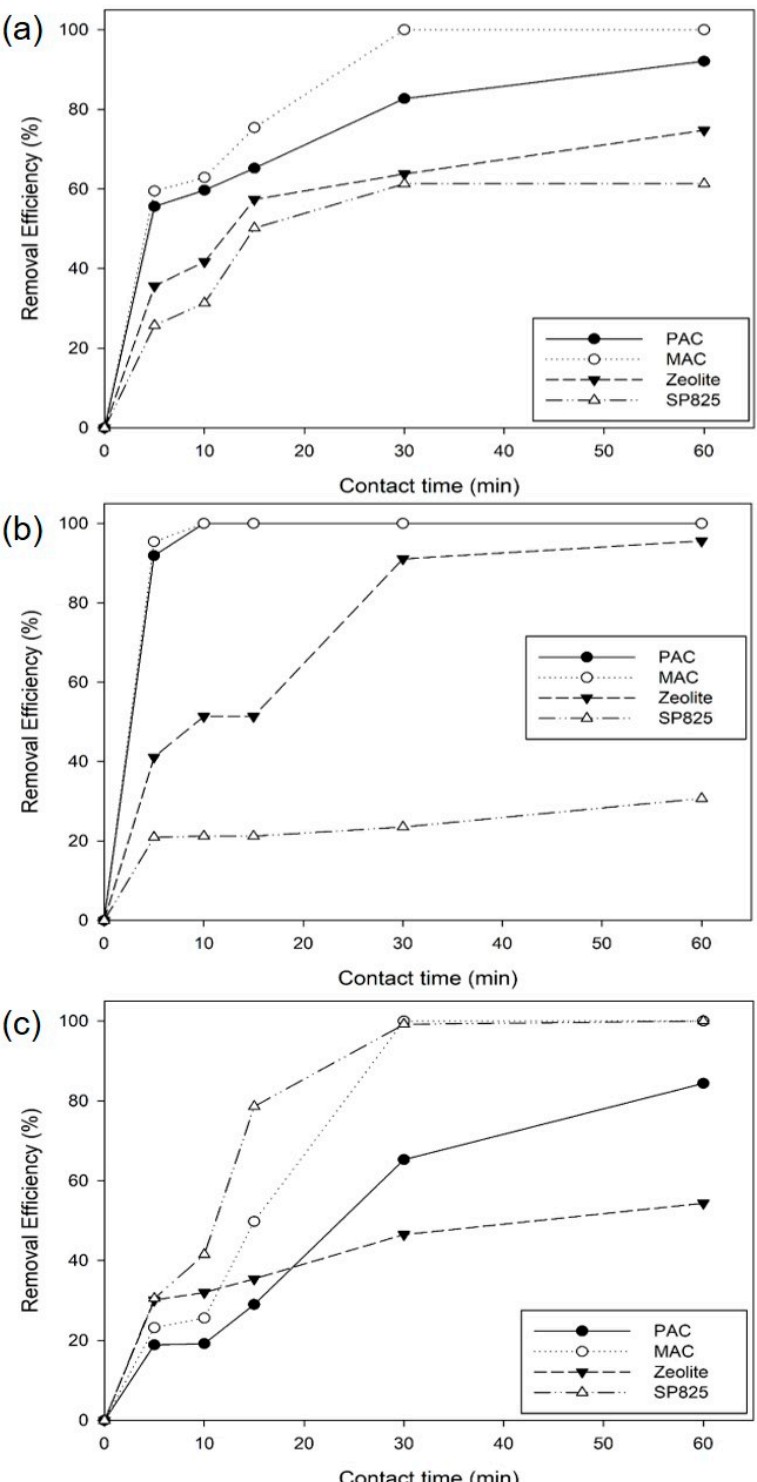

**Figure 2.** Removal efficiency of Sb by contact time ((**a**) Total Sb, (**b**) Sb(III), (**c**) Sb(V)).

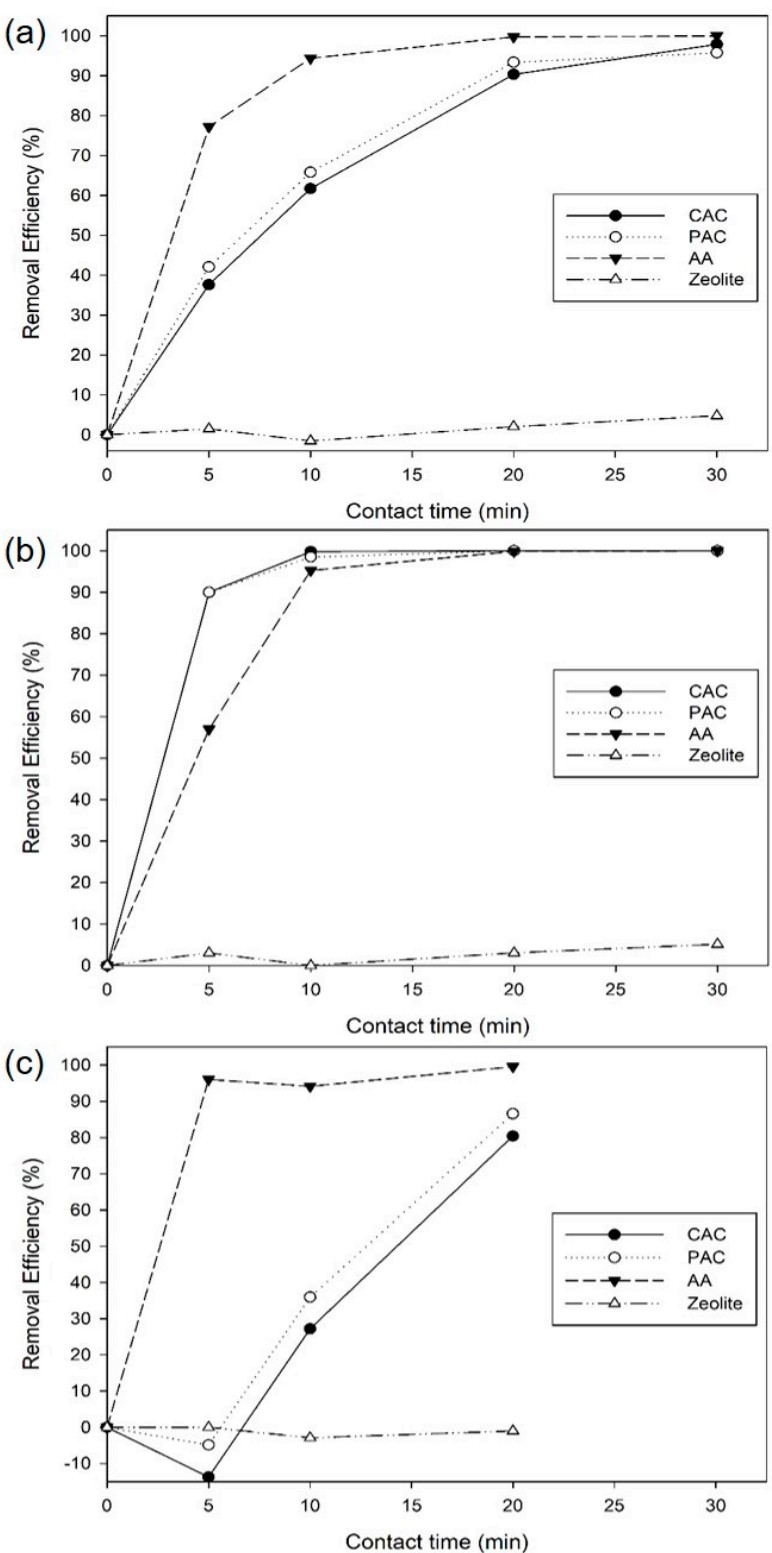

**Figure 3.** Removal efficiency of As by contact time ((**a**) Total As, (**b**) As(III), (**c**) As(V)).

In summary, AA is considered the most efficient substrate for removing arsenic. However, according to recent research results, AA has a positive surface zeta potential and is effective in removing arsenic and antimony in the form of anions in water but has poor removal efficiency for other harmful heavy metals that exist in the form of positive ions [4,16]. Therefore, the optimal adsorbent was selected as CAC, which easily removes highly toxic As(III) through oxidation. The maximum adsorption amount is the highest in

the mixed solution of As(III) and As(V), and the zeta potential is negative, so both cations and anions can be removed. The target concentration for arsenic removal was determined at conditions of LV 1 m/h and SV 2 1/h.

### 3.4. Identification of As(III) Oxidation during the Adsorption Process

In an arsenic removal experiment, the oxidation of As(III) to As(V) by an adsorbent was observed and investigated. Accordingly, the properties of 1 g of CAC which was selected as the optimal adsorbent were analyzed using inductively coupled plasma optical emission spectroscopy (ICP-OES, Optima 8300, PerkinElmer, Waltham, MA, USA) (see Tables 5 and 6). Hence, iron and manganese, which were reported to oxidize arsenic, were confirmed [20,22,24].

**Table 5.** Coal-based activated carbon characteristics before As removal.

| Metal | Concentration (mg) | Metal | Concentration (mg) |
|---|---|---|---|
| As | N.D | Ag | $0.003 \pm 0.001$ |
| Fe | $9.803 \pm 0.005$ | Ni | $0.032 \pm 0.003$ |
| Mg | $0.094 \pm 0.003$ | Pb | N.D |
| Ca | $0.914 \pm 0.010$ | Sr | $0.137 \pm 0.006$ |
| Mn | $0.027 \pm 0.009$ | Sb | N.D |
| Zn | $0.007 \pm 0.001$ | Ti | $0.298 \pm 0.004$ |
| Al | $10.253 \pm 0.057$ | V | N.D |
| Cd | $0.003 \pm 0.001$ | Mo | N.D |
| Na | $1.731 \pm 0.004$ | Sn | N.D |
| Ba | $0.100 \pm 0.008$ | Be | $0.006 \pm 0.001$ |

**Table 6.** Concentrations of As and Fe in coal-based activated carbon after As removal.

| Metal | Concentration (mg) | | |
|---|---|---|---|
| | As(III):As(V) = 2:0 | As(III):As(V) = 1:1 | As(III):As(V) = 0:2 |
| As | $2.496 \pm 0.018$ | $3.891 \pm 0.084$ | $4.235 \pm 0.008$ |
| Fe | $9.900 \pm 0.051$ | $9.893 \pm 0.065$ | $9.843 \pm 0.007$ |

In terms of the proportion of each substance, the amount of iron was the second largest at 41.4%, the amount of manganese was the 13th largest at 0.1%, and the amount of iron was about 360 times higher than that of manganese. Thus, the effect of iron was the main cause of the oxidation phenomenon of arsenic observed for CAC in previous experiments. Additionally, as there was no change in the amount of Fe after arsenic adsorption, it is assumed that no elution of iron existed during adsorption, and that only the oxidation number of Fe changed according to the oxidation reaction of arsenic.

Therefore, 1 L each of 20 mg/L As(III) solution and As(V) solution were prepared, and the pH at this time was neutral pH of about 6.3. Adsorption batch experiments were conducted to investigate the behavior of Fe ions when oxidation reactions did not occur. At this time, stirring was performed at 220 RPM, and samples were collected by filtering 50 mL every 0, 30, 60, 120, 240, 480, and 600 min. and these samples were analyzed using X-ray photoelectron spectroscopy (XPS, K-Alpha+, Thermo Fisher Scientific, Waltham, MA, USA) (see Figure 4).

The As(III) solution initially contained a lot of Fe(II) at 52.1%. Over time, Fe(II) is oxidized to Fe(III) due to the difference in reduction potential, and the amount of Fe(II) increased from 43.8% to a maximum of 48.5%. However, the Fe(II) decreased sharply compared to the increase in Fe(III), from 52.1% to 31.2%. The significant increase in Fe(0) from the initial 4.2% to 20.3% is associated with the reduced Fe(II) through the

oxidation reaction of As(III). Therefore, the amount of Fe(0) rapidly increased in the first 60 min, which was a similar trend to the rapid decrease in the amount of As(III) in the previous experiment. Afterward, the change in Fe(0) gradually decreased and maintained equilibrium for 120 min, which is assumed to be the same as the reduction rate of Fe(II) due to the reduced As(III) and the oxidation rate of Fe(0) due to external factors. After 480 min, the equilibrium of Fe(0) was broken, and its amount decreased from 20.3% to 8.6%. It is believed that as the amount of As(III) decreases, the reaction in which Fe(0) is oxidized to Fe(II) becomes dominant due to external factors such as dissolved oxygen. Iron oxidation in activated carbon favors As(V) over As(III). In the oxidation-based coagulation and precipitation process, operating costs may be reduced by adding spent activated carbon that has reached the break-through point to the preprocessing process instead of the oxidizer [25,26].

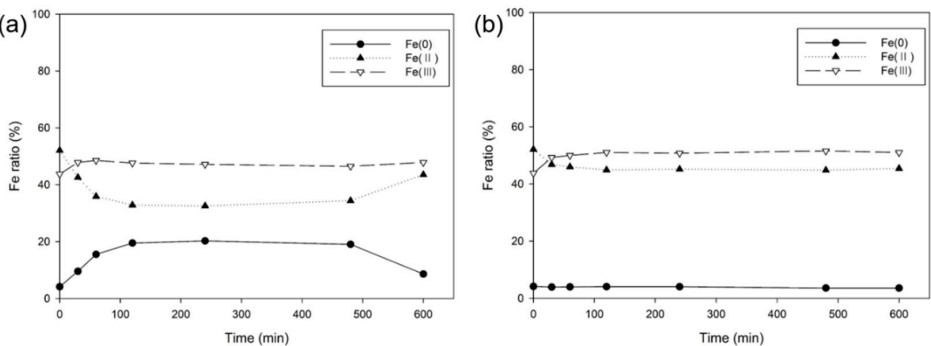

**Figure 4.** Change in iron ions in As solution ((**a**) As(III), (**b**) As(V)).

In the As(V) solution, the amount of Fe(II) was highest at 52.1%; after 30 min, Fe(II) was present at 42.6% and Fe(III) increased from 43.8% to 47.8% as Fe(II) was oxidized to Fe(III). Afterward, Fe(II) decreased from the initial 52.1% to 44.8%, and Fe(III) increased from the initial 43.8% to a maximum of 51.6%. Fe(0) was in an equilibrium state as it was maintained within a constant amount of 4–5% without significant change.

When As(III) was oxidized to As(V), the adsorption and removal efficiency increased as As(III) was oxidized by Fe(II). Accordingly, to use Fe(II) for arsenic removal, the reaction rate order and reaction rate constant between arsenic and iron were obtained through isolation, and the amount and the rate of oxidation of arsenic were confirmed (Table 7). The average reaction rate order of arsenic was 1.04, and that of iron was 1.06, and each ion followed a first-order reaction rate function. Also, the reaction rate constant was initially fast in oxidation but gradually slowed, with an average value of $7.45 \times 10^{-4}$.

**Table 7.** Changes in chemical reaction rate equation constants.

| $V = k \times [As(III)]^a[Fe(II)]^b$ | | | |
|---|---|---|---|
| Time (min) | a (M $\times$ L$^{-1}$) | b (M $\times$ L$^{-1}$) | k (L $\times$ M$^{-1}$s$^{-1}$) |
| 0 | - | - | - |
| 5 | 0.997 | 1.193 | $1.56 \times 10^{-3}$ |
| 10 | 1.007 | 1.103 | $1.39 \times 10^{-3}$ |
| 20 | 1.017 | 1.029 | $1.00 \times 10^{-3}$ |
| 30 | 1.101 | 1.013 | $9.60 \times 10^{-4}$ |
| 60 | 1.104 | 1.007 | $6.45 \times 10^{-4}$ |
| 120 | 1.094 | 0.984 | $4.71 \times 10^{-4}$ |
| 240 | 1.016 | 1.005 | $2.74 \times 10^{-4}$ |
| 360 | 0.994 | 1.029 | $2.09 \times 10^{-4}$ |
| 480 | 0.993 | 1.051 | $1.81 \times 10^{-4}$ |
| 600 | - | 1.14 | - |

*3.5. Confirmation of Removal Characteristics of Metalloid Ions According to Zeta Potential*

To understand the factors affecting the adsorption of metalloids, the surface zeta potential of each adsorbent at pH 7 was measured using a zeta potential analyzer. CAC had a potential of −75 mV at pH 7, 0 mV at pH 2, and −77 mV at pH 13. PAC had a potential of −48 mV at pH 7, 15 mV at pH 2, and −57 mV at pH 13. AA had a potential of 55 mV at pH 7 (pH 3 = 65 mV, pH 13 = −53 mV), zeolite had a potential of −42 mV at pH 7 (pH 2 = −13 mV, pH 13 = −51 mV), while the values for MAC and SP825 were −49 mV and −4 mV, respectively. Antimony and arsenic mainly exist as anions or neutral ions in water, so the removal efficiency was reduced for adsorbents with a negative zeta potential at pH 7. This can be confirmed by the previous experimental results showing that the removal efficiencies of AA and SP825 were particularly high compared to the surface area of the adsorbent. The maximum adsorbed amount for total As at As(III):As(V) = 2:0 was 14.7% higher in efficiency than for CAC, which had the second-best adsorption performance, and As(III):As(V) = 0:2 had an efficiency 89.6% higher than that of CAC. Additionally, zeolite was about 107% more efficient than MAC in processing Sb(V), which mainly exists in the form of $Sb(OH)_6^-$ in water. Adsorbents with similar surface zeta potentials have different removal efficiencies depending on the surface area of each adsorbent, and for others, a higher removal efficiency was observed for zeta potentials closer to a positive charge. Accordingly, the removal of antimony and arsenic was efficient when the surface zeta potential was close to positive charge, considering the versatility and economic efficiency, it would be advantageous to apply CAC or PAC in the actual treatment process.

**4. Conclusions**

This study found the adsorption and removal characteristics of antimony, which has the potential to be carcinogenic, and arsenic, a carcinogen, and selected optimal operating conditions through continuous adsorption experiments. First, an analysis method for total Sb was conducted using voltammetry, which includes preprocessing to oxidize antimony Sb(V) to Sb(III), for which no ion-specific analysis method has been established. Also, removal efficiencies of antimony and arsenic basically showed that the larger the specific surface area of the adsorbent, the greater the adsorption efficiency. However, if there are some polar ion forms such as Sb(V) ($Sb(OH)_6^-$) or As(V) ($H_2AsO_4^-$, $HAsO_4^{2-}$, $AsO_4^{3-}$), it is assumed to be affected by the zeta potential of adsorbent surface, supposing by the experimental results of SP825 or AA, which showed a zeta potential close to neutral. Moreover, the oxidation of As(III) to As(V) during the arsenic adsorption and removal process was caused by Fe(II) in activated carbon, applying that the urgent removal of highly toxic As(III) is required. Based on the results of continuous test, it would be most efficient to use CAC or PAC in the water treatment process, considering versatility and economic efficiency.

**Author Contributions:** Conceptualization, Y.-W.L.; Validation, J.C.; Investigation, S.-H.L.; Data curation, J.C.; Writing—original draft, S.-H.L.; Writing—review & editing, J.C.; Funding acquisition, Y.-W.L. All authors have read and agreed to the published version of the manuscript.

**Funding:** This research was supported by the Ansan Green Environment Center (No. 19-17-03-30-33).

**Data Availability Statement:** Data is contained within the article.

**Conflicts of Interest:** Author Jinwook Chung was employed by the company Samsung Engineering Co., Ltd. The remaining authors declare that the research was conducted in the absence of any commercial or financial relationships that could be construed as a potential conflict of interest.

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
