# Peer review of "Adsorption Removal Characteristics of Hazardous Metalloids (Antimony and Arsenic) According to Their Ionic Properties"

_water, doi:10.3390/w16050767_

Round 1

Reviewer 1 Report

Comments and Suggestions for Authors

1.  Comments and Suggestions for Authors

In this work, the author have investigated on the adsorption and removal characteristics of harmful metals (antimony and arsenic) from the perspective of ionic properties. They have done a nice work. However, in order to make the article better, please consider the following suggestions and a list of suggestions.

1.  From the perspective of ionic properties, studying the adsorption and removal characteristics of antimony and arsenic, different pH values will have an impact on the adsorption performance of antimony and arsenic, and the forms of antimony and arsenic will vary under different pH values. These factors should be studied to make the article more comprehensive and reliable.

2.  Regarding the Langmuir adsorption isotherms and Freundlich adsorption isotherms in the text, please present the specific data in graphs and tables, and also show the formulas used. This makes it easier for readers to understand.

3.  To explain the results of antimony adsorption treatment, data such as the pore size distribution of the selected adsorbent should be presented in a chart. Analyzing different pore size structures can help explain the adsorption results from multiple perspectives.

4.  Please adjust all Figures to ensure that each one is standardized and clear, so that readers can accurately obtain information from Figures.

5.  The Figures of the removal efficiency of As and Sb by contact time should include the points at 0 min, and the experimental conditions should be the same. There are data points taken at 15 min in Figure 2, but there is no data taken at 15 min in Figure 3.

6.  The characterization method of adsorbent characteristics should also be added to the Materials and Methods section. This is beneficial for readers to understand what characterization methods are used to characterize these adsorbents.

7.  In the Results and Discussions, only ASV was used to analyze the total antimony, but total As and As (III) were also measured in Materials and Methods. Add calibration curves for total As and As (III), and the analysis results of total As and As (III) should also be listed in the results and discussion.

8.  “and the amount andrate of oxidation of arsenic were confirmed (Table 5).” This should be a reference to Table 6.

9.  Many experimental results in the conclusion section can enrich the conclusion section, but can be further simplified to highlight the adsorption and removal characteristics of antimony and arsenic.

10. The summary of the adsorption and removal characteristics of antimony and arsenic is not concise enough. Summarize the adsorption and removal characteristics of antimony and arsenic in a concise manner.

11. All equations used should be inserted into the main text. And annotate the equation.

12. Please thoroughly proofread and delete or modify sentences with repetitive meanings to refine the article, correct any grammar, formatting, typing errors, or clumsy wording. All Figures, Schemes, and Tables should be inserted into the main text. Please carefully check each title, please fill in according to the requirements of the journal.

13. Journal references must cite the full title of the paper, page range or article number, and digital object identifier (DOI) where available. Please fill in according to the requirements of the journal.

Comments on the Quality of English Language

Minor editing of English language required

Author Response

Reviewer #1:

In this work, the author have investigated on the adsorption and removal characteristics of harmful metals (antimony and arsenic) from the perspective of ionic properties. They have done a nice work. However, in order to make the article better, please consider the following suggestions and a list of suggestions.

  1. From the perspective of ionic properties, studying the adsorption and removal characteristics of antimony and arsenic, different pH values will have an impact on the adsorption performance of antimony and arsenic, and the forms of antimony and arsenic will vary under different pH values. These factors should be studied to make the article more comprehensive and reliable.

This study aimed to provide results that can be directly applied in actual industrial settings. Accordingly, the conditions were carried out in a way that simulated the wastewater treatment process in an actual industrial site. In actual sites, wastewater is treated by maintaining neutral pH, so the experiment was conducted accordingly.

  1. Regarding the Langmuir adsorption isotherms and Freundlich adsorption isotherms in the text, please present the specific data in graphs and tables, and also show the formulas used. This makes it easier for readers to understand.

Data using Langmuir and Freundlich adsorption isotherms are summarized in Tables 2 and 4, respectively. The related equations are demonstrated in “session 3.1”

  1. To explain the results of antimony adsorption treatment, data such as the pore size distribution of the selected adsorbent should be presented in a chart. Analyzing different pore size structures can help explain the adsorption results from multiple perspectives.

In this study, only BET analysis was performed, and the pore size distribution you mentioned was not performed during the experiment, so it cannot be presented in this paper.

  1. Please adjust all Figures to ensure that each one is standardized and clear, so that readers can accurately obtain information from Figures.

According to referee’s comment, we modified all figures to be standardized and clear as possible (see Figure 1, 2 & 3)

  1. The Figures of the removal efficiency of As and Sb by contact time should include the points at 0 min, and the experimental conditions should be the same. There are data points taken at 15 min in Figure 2, but there is no data taken at 15 min in Figure 3.

We added 0 min to Figure 2 and Figure 3.

  1. The characterization method of adsorbent characteristics should also be added to the Materials and Methods section. This is beneficial for readers to understand what characterization methods are used to characterize these adsorbents.

Adsorbent characteristics were organized in Table 1.

“The physical properties provided by the manufacturer of the selected adsorbent are summarized in Table 1.”

Also, analysis methods are summarized as follows:

“Before the experiment, the specific surface area of each adsorbent was analyzed using a Brunauer Emmett Teller Analyzer (MicrotracBEL Corp., BELSORP MAX X, Japan). Also, the zeta potential was analyzed with a zeta potential analyzer (Brookhaven Instruments Corporation, NanoBrook ZetaPALS Potential Analyzer, USA) at pH 7.”

  1. In the Results and Discussions, only ASV was used to analyze the total antimony, but total As and As (III) were also measured in Materials and Methods. Add calibration curves for total As and As (III), and the analysis results of total As and As (III) should also be listed in the results and discussion.

We added calibration curves to Figure 1.

Also, analysis results of total As and As (III) should also be listed in the results and discussion.

“Sb(Ⅲ), As(Ⅲ), and total As were analyzed using the measurement method distributed by the ASV manufacturer, and the R2 was Sb(Ⅲ): 0.9693, As(Ⅲ): 0.9980, and total As: was found to be 0.9925 (Fig. 1).”

  1. “and the amount and rate of oxidation of arsenic were confirmed (Table 5).” This should be a reference to Table 6.

Accordingly, we corrected the number of Table.

“Accordingly, to use Fe(II) for arsenic removal, the reaction rate order and reaction rate constant between arsenic and iron were obtained through isolation, and the amount and the rate of oxidation of arsenic were confirmed (Table 7).”

  1. Many experimental results in the conclusion section can enrich the conclusion section, but can be further simplified to highlight the adsorption and removal characteristics of antimony and arsenic.

The conclusion section has been summarized and modified to emphasize the adsorption characteristics.

“This study found the adsorption and removal characteristics of antimony, which has the potential to be carcinogenic, and arsenic, a carcinogen, and selected optimal operating conditions through continuous adsorption experiments. First, an analysis method for total Sb was conducted using voltammetry, which includes preprocessing to oxidize antimony Sb(V) to Sb(Ⅲ), for which no ion-specific analysis method has been established. Also, removal efficiencies of antimony and arsenic basically showed that the larger the specific surface area of the adsorbent, the greater the adsorption efficiency. However, if there are some polar ion forms such as Sb(V) (Sb(OH)6-) or As(V) (H2AsO4-, HAsO42-, AsO43-), it is assumed to be affected by the zeta potential of adsorbent surface, supposing by the experimental results of SP825 or AA, which showed a zeta potential close to neutral. Moreover, the oxidation of As(Ⅲ) to As(Ⅴ) during the arsenic adsorption and removal process was caused by Fe(Ⅱ) in activated carbon, applying that the urgent removal of highly toxic As(Ⅲ) is required. Based on the results of continuous test, it would be most efficient to use CAC or PAC in the water treatment process, considering versatility and economic efficiency.”

  1. The summary of the adsorption and removal characteristics of antimony and arsenic is not concise enough. Summarize the adsorption and removal characteristics of antimony and arsenic in a concise manner.

The conclusion section has been summarized and modified to emphasize the adsorption characteristics.

  1. All equations used should be inserted into the main text. And annotate the equation.

The adsorption isotherm equation used to calculate the maximum capacity of adsorption is summarized and annotated the related equations to the text.

  1. Please thoroughly proofread and delete or modify sentences with repetitive meanings to refine the article, correct any grammar, formatting, typing errors, or clumsy wording. All Figures, Schemes, and Tables should be inserted into the main text. Please carefully check each title, please fill in according to the requirements of the journal.

Grammar, formatting, typos and clumsy wording have been corrected throughout the whole text. In addition, all figures and tables were inserted into the main text.

  1. Journal references must cite the full title of the paper, page range or article number, and digital object identifier (DOI) where available. Please fill in according to the requirements of the journal.

As you mentioned, DOIs were added to all possible references, and page range and article numbers were all checked and rearranged according to the requirements of the journal.

Reviewer 2 Report

Comments and Suggestions for Authors

Dear Authors, please, see PDF file attached.

Author Response

Reviewer #2:

Herein, I submit my comments for submission of your manuscript entitled: “Adsorption Removal Characteristics of Hazardous Metalloids (Antimony and Arsenic) According to Their Ionic Properties”.

I consider the search for new ways of decontamination of the water mass from toxic elements to be very important. There are many techniques and ways to remove Sb and As. Claims for removing As and Sb and at the same time polishing water quality are almost always related to the price and effectiveness of the given material, process or method. Carbon-based materials are a very reliable and cheaper way to achieve the desired result. Therefore, I consider research in this area to be current and important.

I see the biggest shortcoming of the entire manuscript in the description of methods (procedures), which are insufficient. As a result, it is not possible to make a professional assessment of this work of high quality. I miss the logical continuity of the method of obtaining the result and the result itself. The authors do not talk about uncertainties at all, and they are not commented or evaluated in any way in the manuscript. This does not give a very good picture of the scientific soundness of the manuscript. The whole work is thus not transparent.

For the reasons stated above, I decided to give the authors a chance to correct the manuscript. I ask the authors to edit the manuscript according to my comments in the table.

  1. P1 L16, …in case you mean chemical forms other than cations, it is necessary to write it here...e.g.: anionic form of Sb(V) etc…

We added the ionic form of all substances.

“Sb(III) and Sb(V) ions exist as Sb(OH)3 and Sb(OH)6-, respectively, at pH 7, while arsenic exists in a total of eight neutral molecules or anions (i.e., H3AsO3, H3AsO4, H2AsO3-, H2AsO4-, HAsO32-, HAsO42-, AsO33-, AsO43-) in water depending on temperature and pH (Chen et al., 1994; Ha, 1978).”

  1. P1 L22, the last word is far from the others

We checked the double space and corrected this.

  1. P2 L78 – L80, prepared from

We replaced “prepared by” by “prepared from”.

  1. P2 L85 – L86 + L90 – L91, 1) How sure are you about the speciation of Sb (and As as well)? Please support this statement with a reference to the literature or your own findings.

2) Is the subtraction of concentrations a standard procedure? 3) It is not clear, what chemical did you used for “total Sb”. Please make it clear. And Ad4) how did you prepared calibration curves for “total Sb” and “Sb(III)” – what is de difference? the same questions apply to As

The speciation of Sb and As was determined through literature review. Especially, In the case of Sb(V), because there was no existing analysis method, a new analysis method was described in “session 3.1”.

“Considering the form of Sb(V) in water depending on pH, previous research results showed that Sb(V) exists in the form of Sb(Ⅲ) in a highly concentrated HCl solution (5 M or more) (Whitney and Davidson, 1949). However, if the sample is preprocessed with HCl, the mercury film coated on the working electrode of the ASV dissolves, pre-venting measurements. To prevent the mercury film from dissolving, the sample pre-processed with a 5 M HCl solution was re-neutralized into a basic solution. However, as the pH became neutral, Sb(Ⅲ) was oxidized back to Sb(V).”

The analysis of Sb(Ⅲ) and As were conducted according to the manual of Modern Water Co., an ASV manufacturer.

“Sb(Ⅲ), As(Ⅲ), and total As were analyzed using the measurement method distributed by the ASV manufacturer, and the R2 was Sb(Ⅲ): 0.9693, As(Ⅲ): 0.9980, and total As: was found to be 0.9925 (Fig. 1).”

  1. P3 L98, This statement belongs in the results section. Where are the R2 values for Sb?

Accordingly, we moved the mentioned sentence and added R2 values in the text.

“Sb(Ⅲ), As(Ⅲ), and total As were analyzed using the measurement method distributed by the ASV manufacturer, and the R2 was Sb(Ⅲ): 0.9693, As(Ⅲ): 0.9980, and total As: was found to be 0.9925 (Fig. 1).”

  1. P3 L106- L107, L112 – L113, What does it mean? What should be the zeta potential for the given material to be satisfactory in your opinion?

We added the meaning of natural and positive zeta potential.

“Zeolite (Dongsin, 0.6–2.4 mm, Korea) had a neutral zeta potential was chosen as the control due to its smallest least adsorption effect. Moreover, MAC was made by react-ing a previous PAC for 120 minutes at 900°C while flowing 1000 mL/min of CO2 gas to improve adsorption efficiency (Ha, 1978; US EPA, 2021). For arsenic, coal-based acti-vated carbon (CAC Jeilcarbontec, 0.6-2.4 mm, Korea) was selected as an adsorbent to replace the commercial adsorbents PAC and MAC. Activated alumina (AA, BASF, 2 mm, Germany) had a positive zeta potential was selected considering zeta potential, and Zeolite (Dongsin, 0.6–2.4 mm, Korea) was selected as a control adsorbent (Ha, 1978; Singh & Pant, 2004; US EPA, 2021).”

  1. P3 L108 – L109, 1) In what did you prepared MAC? 2) cc/ml – please use SI units.

We replaced “prepared” and “cc” by  “made” and “mL"

  1. P3 L114 – L119, This statement belongs to the section with the evaluation of the results.

This statement indicates the justification for selecting the size of the adsorbent. Accordingly, in order to prevent misunderstanding regarding the content of the result, we revised the mentioned sentence.

“Considering this, the diameter range was 0.6–2.4 mm, which is mainly used for water processing.”

  1. P3 L119 – L120, Grammar. Incomprehensible statement.

We revised to avoid the ambiguity.

“In the case of antimony, 30 mL of Sb(III) and Sb(V) solutions with a concentration of 0.2 mg/L each were added to the conical tube, and 20 mg of each adsorbent was add-ed. Afterwards, it was stirred for 10, 30, 60, 120, and 240 min and then filtered to collect samples.”

  1. P3 L122, Samples of what (and how much) were collected?

We revised to avoid the ambiguity.

“In the case of antimony, 30 mL of Sb(III) and Sb(V) solutions with a concentration of 0.2 mg/L each were added to the conical tube, and 20 mg of each adsorbent was add-ed. Afterwards, it was stirred for 10, 30, 60, 120, and 240 min and then filtered to collect samples.”

  1. P3 L123 – L124, Simply wetting the sorbent is not sufficient. Due to its physical nature, it is impossible to get air out of pores that are on the "micro" and "meso" scale... In this case, it is necessary to reduce the air pressure above the suspension!

The reason for removing air from the pores is that in experiments that require continuous stirring, if air is still contained within the pores, the adsorbent will continue to float, making it difficult to properly stir. Therefore, we added the sentence.

“The air in the adsorbent pores was removed to solve the problem of the adsorbent floating during the experiment.”

  1. P3 L125, Incomprehensible statement. What kind of device is that?

The jar tester is a commercially available equipment for jar testing. It is equipped with 6 constant speed stirring devices, allowing experiments to be conducted under the same test conditions. To avoid confusion, we added model name and manufacturing company  after jar tester.

“As solution and then stirred using a jar tester (C-JY-H, Vision Lab Science, Korea) at 120 rpm for 1,440 min to ensure complete mixing of the adsorbent and solution.”

  1. P3 L126 – L127, Incomprehensible statement. What are those ratios? As(III) : As(V)? As(III) : solid phase?

We replaced “The ratios of As(Ⅲ) and As(V)” by “The ratios between As(Ⅲ) and As(V)” to avoid the ambiguity.

  1. P3 L127, Incomprehensible statement. On the basis of which data did you come to the conclusion that "removal efficiency was confirmed"?

We replaced “confirmed” by “found”.

  1. P3 L133 – L134 d, V = π·d²·h = 5.4428 L ??

V = π·(1.5 cm/2)²·170 cm = 300.3 cm3 = 300.3 mL

  1. P3 L143, What does it mean?

We replaced “surface unit velocity” by “surface velocity” to avoid the ambiguity,

  1. P3 L144 – L145, P4 L152 – L153, Does that mean you put more materials in every single column?

We revised to avoid the ambiguity.

“Afterwards, the amount of adsorbent filling was changed to increase the contact time. At this time, the change in filling amount started from 33 cm (LV 1, SV 3), which is the adsorbent height in the existing flow rate change test, and was applied up to 100 cm (LV 1, SV 1).”

  1. P4 L172, 1) This should be in 2.1 Chemicals. 2) Blue colour

According to referee’s comment. We revised.

“As reagents for analysis, HCl (Fluka, Trace Select®, Switzerland), L-ascorbic acid (Sig-ma Al-drich, ACS reagent, USA), and KOH (Sigma Aldrich, 99.99% trace metals basis, USA) were used.”

  1. P4 L156 – L157, What previous studies? Reference? E.g. DOI 10.1016/j.talanta.2022.123578 ?

Unfortunately, we could not find any previous studies on total antimony analysis using ASV.

  1. P4 L162, The speciation should be supported by Eh-pH diagram or by your own research.

This speciation was supported by Eh-pH diagram.  

  1. P4 L178, Metalloid adsorption removal characteristics

Accordingly, we revied the sub title.

“3.1. Metalloids analysis using ASV”

  1. P4 L184, Are you writing about "confirmation" - does that mean that the results already exist? Please, explain.

We replaced “confirm” by “find”

  1. P4 L185 – L186, not quite a scientific phrase…

According to referee’s comment, we revised the mentioned sentence.

“After writing and comparing the Langmuir adsorption isotherm (1) and the Freundlich equation (2), a more appropriate adsorption isotherm was determined.”

  1. P4 L188, P5 L240, Grammar.

We replaced “an” by “a”.

  1. P4 L199, Reference to a figure is missing.

Since the specific surface area of each adsorbent was explained earlier in the manuscript about the adsorption performance of each adsorbent, no additional graphs were added.

  1. P5 L242, 1) Not a very well chosen chapter title. Proposal for change: column experiments, break-through experiments, continuous adsorption tests, etc. 2) Please, delete the dash.

Accordingly, we revised the sub title and deleted the mentioned sentence.

  1. P5 L242, Please also include the experimental problems. For sure, you must have had an issue with aeration of the column for such long-lasting experiments. This happens normally. Please comment

Since there was an aeration problem in the column, to minimize this, the adsorbent soaked in distilled water for 1 day before the experiment was mixed with distilled water and packed in a way that it flowed down together. At this time, the air contained in the large pores is released, making packing at the bottom of the column easier. In addition, distilled water was flowed through each column for one day and an appropriate amount of vibration was applied to ensure that there were no gaps and air bubbles rose to the top.

  1. P6 L280 – L281, A bit strange logic of consecutive statements…

We revised to avoid ambiguity.

“Accordingly, the properties of 1 g of CAC which was selected as the optimal adsorbent were analyzed using Inductively Coupled Plasma Optical Emission Spectroscopy (ICP-OES, Optima 8300, PerkinElmer, USA) (see Tables 5 and 6).”

  1. P6 L292 – L293, 1) What adsorption experiments? In a column or batch type? 2) Again, as above. You are claiming ,,confirmation”… Does it mean that you are doing experiments that have been already done (by someone else)? - anywhere else in the manuscript. The word ,,confirmation” it gives the feeling that your work only confirms something but does not bring anything new. Please try to avoid this word in your work.

Accordingly, we replaced “adsorption experiments” by “adsorption batch experiments” and avoided “confirmation” wording throughout the whole manuscript.

  1. P6 L292, How were these solutions prepared? Concentration, pH, etc.?

We revised this sentence to help the readers understanding.

“1 L each of 20 mg/L As(III) solution and As(V) solution were prepared, and the pH at this time was neutral pH of about 6.3”

  1. P6 L294, Without a measurement procedure, it is not possible to review this section. Please, writhe here the measuring procedure. Some things are not clear here, e.g. were the samples measured wet or were they dried after sorption? What was the pH? etc.

We revised this sentence to help the readers understanding.

“Adsorption batch experiments were conducted to investigate the behavior of Fe ions when oxidation reactions did not occur. At this time, stirring was performed at 220 RPM, and samples were collected by filtering 50 mL every 0, 30, 60, 120, 240, 480, and 600 minutes. and these samples were analyzed using X-ray Photoelectron Spectroscopy (XPS, K-Alpha+, Thermo Fisher Scient, USA) (see Fig. 4).”

  1. P6 L296 – L298, It is not clear here what is your result and what is the result found in the literature.

We deleted the mentioned reference.

  1. P6 L302, Incomprehensible statement

We revised to avoid the ambiguity.

“Therefore, the amount of Fe(0) rapidly increased in the first 60 minutes, which was a similar trend to the rapid decrease in the amount of As(Ⅲ) in the previous experiment.”

  1. P7 L308, What factors?

We revised to avoid the ambiguity.

“It is believed that as the amount of As(III) decreases, the reaction in which Fe(0) is oxidized to Fe(II) becomes dominant due to external factors such as dissolved oxygen.”

  1. P7 L321, and the rate

We corrected this.

  1. P7 L337, Grammar. “adsorbed amount”

We replaced “maximum adsorption amount” by “maximum adsorbed amount”.

  1. P7 L338, second best

We corrected this.

  1. P7 L348, For the reason already stated above, please indicate not "confirmation" but what new your results bring. What they add to the issue in question.

Accordingly, we avoided “confirmation” wording throughout the whole manuscript.

  1. P8 L356, Incomprehensible statement

The conclusion section has been summarized and modified to emphasize the adsorption characteristics.

“This study found the adsorption and removal characteristics of antimony, which has the potential to be carcinogenic, and arsenic, a carcinogen, and selected optimal operating conditions through continuous adsorption experiments. First, an analysis method for total Sb was conducted using voltammetry, which includes preprocessing to oxidize antimony Sb(V) to Sb(Ⅲ), for which no ion-specific analysis method has been established. Also, removal efficiencies of antimony and arsenic basically showed that the larger the specific surface area of the adsorbent, the greater the adsorption efficiency. However, if there are some polar ion forms such as Sb(V) (Sb(OH)6-) or As(V) (H2AsO4-, HAsO42-, AsO43-), it is assumed to be affected by the zeta potential of adsorbent surface, supposing by the experimental results of SP825 or AA, which showed a zeta potential close to neutral. Moreover, the oxidation of As(Ⅲ) to As(Ⅴ) during the arsenic adsorption and removal process was caused by Fe(Ⅱ) in activated carbon, applying that the urgent removal of highly toxic As(Ⅲ) is required. Based on the results of continuous test, it would be most efficient to use CAC or PAC in the water treatment process, considering versatility and economic efficiency.”

Round 2

Reviewer 1 Report

Comments and Suggestions for Authors

The manuscript has been sufficiently revised and can be accepted in its present form.

Author Response

Because Reviewer 1's coments is none, we did not submit any response.

Reviewer 2 Report

Comments and Suggestions for Authors

Dear Authors, you will find my comments attached.

Author Response

Feb 28, 2024

Manuscript ID: water-2879599

Title: Adsorption Removal Characteristics of Hazardous Metalloids (Antimony and Arsenic) According to Their Ionic Properties
Author(s): Seung-Hun Lee, Jinwook Chung, Yong-Woo Lee

Dear Sir/Madam

This letter addresses revisions to manuscript Ms. water-2879599, which was submitted for publication. The manuscript has been modified to incorporate all the comments of the reviewers. Attached is a list of specific comments and our responses. The reviewers’ comments are in black, followed by the authors’ responses in blue. The changes that were made in response to the comments are highlighted in the text in yellow.

Please contact me if you have any questions or immediate concerns.

Sincerely,

Yong-Woo Lee, Ph.D.

Professor

Department of Chemical and Molecular Engineering, Hanyang University,

1271 Sa 1-Dong, Sangnok-Gu, Ansan, Gyeonggi-Do 426-791, Republic of Korea

Tel: +82-31-400-5508

Fax: +82-31-407-3863

E-mail: [email protected]
Reviewer #2:

Dear Authors,

thank you for the review of your paper. I ask the authors to make changes according to my suggestions in the table.

  1. P3 L108, 0.4 mm

- We changed “0.6-.4 mm” by “0.6-2.4 mm”.

“For antimony, palm-based activated carbon (PAC, Jeilcarbontec, 0.6–2.4 mm, Korea) was selected as a commonly used commercial adsorbent”

  1. P4 L192, red line

- We changed “HCl-and” by “HCl and”.

“For the final preprocessing, 45 mL of 5 M HCl and 1.4 g of L-ascorbic acid were added to 10 mL of the sample and left for 10 minutes”

  1. P7 L272, Korean character

- We changed “Compared with Langmuir adsorption isotherm (1)과 Freundlich adsorption isotherm” by “Compared with Langmuir adsorption isotherm (1) and Freundlich adsorption isotherm”.

“Compared with Langmuir adsorption isotherm (1) and Freundlich adsorption isotherm, a more appropriate adsorption isotherm was determined.”
